# Deep Generalized Method of Moments
# for Instrumental Variable Analysis

**Andrew Bennett**[*]
Cornell University
awb222@cornell.edu

**Nathan Kallus**[*]
Cornell University
kallus@cornell.edu

**Tobias Schnabel**[*]
Microsoft Research
tbs49@cornell.edu

## Abstract

Instrumental variable analysis is a powerful tool for estimating causal effects when randomization or full control of confounders is not possible. The application of standard methods such as 2SLS, GMM, and more recent variants are significantly impeded when the causal effects are complex, the instruments are high-dimensional, and/or the treatment is high-dimensional. In this paper, we propose the DeepGMM algorithm to overcome this. Our algorithm is based on a new variational reformulation of GMM with optimal inverse-covariance weighting that allows us to efficiently control very many moment conditions. We further develop practical techniques for optimization and model selection that make it particularly successful in practice. Our algorithm is also computationally tractable and can handle large-scale datasets. Numerical results show our algorithm matches the performance of the best tuned methods in standard settings and continues to work in high-dimensional settings where even recent methods break.

## 1 Introduction

Unlike standard supervised learning that models correlations, causal inference seeks to predict the effect of counterfactual interventions not seen in the data. For example, when wanting to estimate the effect of adherence to a prescription of $\beta$-blockers on the prevention of heart disease, supervised learning may overestimate the true effect because good adherence is also strongly correlated with health consciousness and therefore with good heart health [13]. Figure 1 shows a simple example of this type and demonstrates how a standard neural network (in blue) fails to correctly estimate the true treatment response curve (in orange) in a toy example. The issue is that standard supervised learning assumes that the residual in the response from the prediction of interest is independent of the features.

One approach to account for this is by adjusting for all confounding factors that cause the dependence, such as via matching [24, 33] or regression, potentially using neural networks [23, 25, 34]. However, this requires that we actually observe *all* confounders so that treatment is as-if random after conditioning on observables. This would mean that in the $\beta$-blocker example, we would need to perfectly measure *all* latent factors that determine both an individual's adherence decision and their general healthfulness which is often not possible in practice.

Instrumental variables (IVs) provide an alternative approach to causal-effect identification. If we can find a latent experiment in another variable (the instrument) that influences the treatment (*i.e.*, is relevant) and does not directly affect the outcome (*i.e.*, satisfies exclusion), then we can use this to infer causal effects [3]. In the $\beta$-blocker example [13], the authors used co-pay cost as an IV. Because they enable analyzing natural experiments under mild assumptions, IVs have been one of the most widely used tools for empirical research in a variety of fields [2]. An important direction of research

---

[*]Alphabetical order.

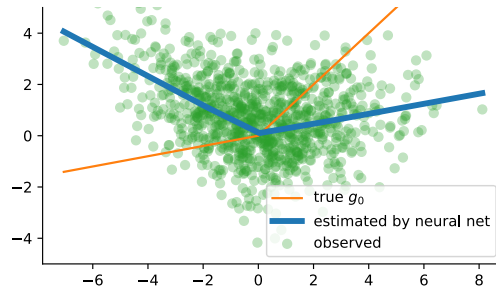

Figure 1: A toy example in which standard supervised learning fails to identify the true response function $g_0(X) = \max(\frac{X}{5}, X)$. Data was generated via $Y = g_0(X) - 2\epsilon + \eta$, $X = Z + 2\epsilon$. All other variables are standard normal.

for IV analysis is to develop methods that can effectively handle complex causal relationships and complex variables like images that necessitate more flexible models like neural networks [21, 28].

In this paper, we tackle this through a new method called DeepGMM that builds upon the optimally-weighted Generalized Method of Moments (GMM) [17], a widely popular method in econometrics that uses the moment conditions implied by the IV model to efficiently estimate causal parameters. Leveraging a new variational reformulation of the efficient GMM with optimal weights, we develop a flexible framework, DeepGMM, for doing IV estimation with neural networks. In contrast to existing approaches, DeepGMM is suited for high-dimensional treatments $X$ and instruments $Z$, as well as for complex causal and interaction effects. DeepGMM is given by the solution to a smooth game between a prediction function and critic function. We prove that approximate equilibria provide consistent estimates of the true causal parameters. We find these equilibria using optimistic gradient descent algorithms for smooth game play [15], and give practical guidance on how to choose the parameters of our algorithm and do model validation. In our empirical evaluation, we demonstrate that DeepGMM's performance is on par or superior to a large number of existing approaches in standard benchmarks and continues to work in high-dimensional settings where other methods fail.

## 2   Setup and Notation

We assume that our data is generated by

$$Y = g_0(X) + \epsilon, \tag{1}$$

where the residual $\epsilon$ has zero mean and finite variance, *i.e.*, $\mathbb{E}[\epsilon] = 0$ and $\mathbb{E}[\epsilon^2] < \infty$. However, different to standard supervised learning, we allow for the residual $\epsilon$ and $X$ to be correlated, $\mathbb{E}[\epsilon \mid X] \neq 0$, *i.e.*, $X$ can be endogenous, and therefore $g_0(X) \neq \mathbb{E}[Y \mid X]$. We also assume that we have access to an instrument $Z$ satisfying

$$\mathbb{E}[\epsilon \mid Z] = 0. \tag{2}$$

Moreover, $Z$ should be relevant, *i.e.* $\mathbb{P}(X \mid Z) \neq \mathbb{P}(X)$. Our goal is to identify the causal response function $g_0(\cdot)$ from a parametrized family of functions $G = \{g(\cdot; \theta) : \theta \in \Theta\}$. Examples are linear functions $g(x; \theta) = \theta^T \phi(x)$, neural networks where $\theta$ represent weights, and non-parametric classes with infinite-dimensional $\theta$. For convenience, let $\theta_0 \in \Theta$ be such that $g_0(\cdot) = g(\cdot; \theta_0)$. Throughout, we measure the performance of an estimated response function $\hat{g}$ by its MSE against the true $g_0$.

Note that if we additionally have some exogenous context variables $L$, the standard way to model this using Eq. (1) is to include them both in $X$ and in $Z$ as $X = (X', L)$ and $Z = (Z', L)$, where $X'$ is the endogenous variable and $Z'$ is an IV for it. In the $\beta$-blocker example, if we were interested in the heterogeneity of the effect of adherence over demographics, $X$ would include both adherence and demographics whereas $Z$ would include both co-payment and demographics.

## 2.1 Existing methods for IV estimation

**Two-stage methods.** One strategy to identifying $g_0$ is based on noting that Eq. (2) implies

$$\mathbb{E}\left[Y \mid Z\right] = \mathbb{E}\left[g_0(X) \mid Z\right] = \int g_0(x) d\mathbb{P}\left(X = x \mid Z\right). \tag{3}$$

If we let $g(x; \theta) = \theta^T \phi(x)$ this becomes $\mathbb{E}\left[Y \mid Z\right] = \theta_0^T \mathbb{E}\left[\phi(X) \mid Z\right]$. The two-stage least squares (2SLS) method [3, §4.1.1] first fits $\mathbb{E}\left[\phi(X) \mid Z\right]$ by least-squares regression of $\phi(X)$ on $Z$ (with $Z$ possibly transformed) and then estimates $\hat{\theta}^{2\text{SLS}}$ as the coefficient in the regression of $Y$ on $\mathbb{E}\left[\phi(X) \mid Z\right]$. This, however, fails when one does not know a sufficient basis $\phi(x)$ for $g(x, \theta_0)$. [14, 29] propose non-parametric methods for expanding such a basis but such approaches are limited to low-dimensional settings. [21] instead propose DeepIV, which estimates the conditional density $\mathbb{P}\left(X = x \mid Z\right)$ by flexible neural-network-parametrized Gaussian mixtures. This may be limited in settings with high-dimensional $X$ and can suffer from the non-orthogonality of MLE under any misspecification, known as the "forbidden regression" issue [3, §4.6.1] (see Section 5 for discussion).

**Moment methods.** The generalized method of moments (GMM) instead leverages the moment conditions satisfied by $\theta_0$. Given functions $f_1, \ldots, f_m$, Eq. (2) implies $\mathbb{E}\left[f_j(Z)\epsilon\right] = 0$, giving us

$$\psi(f_1; \theta_0) = \cdots = \psi(f_m; \theta_0) = 0, \quad \text{where} \quad \psi(f; \theta) = \mathbb{E}\left[f(Z)(Y - g(X; \theta))\right]. \tag{4}$$

A usual assumption when using GMM is that the $m$ moment conditions in Eq. (4) are sufficient to uniquely pin down (identify) $\theta_0$.[2] To estimate $\theta_0$, GMM considers these moments' empirical counterparts, $\psi_n(f; \theta) = \frac{1}{n} \sum_{i=1}^n f(Z_i)(Y_i - g(X_i; \theta))$, and seeks to make all of them small simultaneously, measured by their Euclidean norm $\|v\|^2 = v^T v$:

$$\hat{\theta}^{\text{GMM}} \in \underset{\theta \in \Theta}{\text{argmin}} \left\|(\psi_n(f_1; \theta), \ldots, \psi_n(f_m; \theta))\right\|^2. \tag{5}$$

Other vector norms are possible. [28] propose using $\|v\|_\infty$ and solving the optimization with no-regret learning along with an intermittent jitter to moment conditions in a framework they call AGMM (see Section 5 for discussion).

However, when there are many moments (many $f_j$), using *any* unweighted vector norm can lead to significant inefficiencies, as we may be wasting modeling resources to make less relevant or duplicate moment conditions small. The *optimal* combination of moment conditions, yielding minimal variance estimates is in fact given by weighting them by their inverse covariance, and it is sufficient to consistently estimate this covariance. In particular, a celebrated result [17] shows that (with finitely-many moments), using the following norm in Eq. (5) will yield *minimal* asymptotic variance (efficiency) for any consistent estimate $\tilde{\theta}$ of $\theta_0$:

$$\|v\|_{\tilde{\theta}}^2 = v^T C_{\tilde{\theta}}^{-1} v, \quad \text{where} \quad [C_\theta]_{jk} = \frac{1}{n} \sum_{i=1}^n f_j(Z_i) f_k(Z_i)(Y_i - g(X_i; \theta))^2. \tag{6}$$

Examples of this are the two-step, iterative, and continuously updating GMM estimators [20]. We generically refer to the GMM estimator given in Eq. (5) using the norm given in Eq. (6) as *optimally-weighted GMM* (OWGMM), or $\hat{\theta}^{\text{OWGMM}}$.

**Failure of (OW)GMM with Many Moment Conditions.** When $g(x; \theta)$ is a flexible model such as a high-capacity neural network, many – possibly infinitely many – moment conditions may be needed to identify $\theta_0$. However, GMM and OWGMM algorithms fail when we use too many moment conditions. On the one hand, one-step GMM (*i.e.*, Eq. (5) with $\|v\| = \|v\|_p$, $p \in [1, \infty]$) is saddled with the inefficiency of trying to impossibly control many equally-weighted moments: at the extreme, if we let $f_1, \ldots$ be all functions of $Z$ with unit square integral, one-step GMM is simply equivalent to the *non-causal* least-squares regression of $Y$ on $X$. We discuss this further in Appendix C. On the other hand, we also cannot hope to learn the optimal weighting: the matrix $C_{\tilde{\theta}}$ in Eq. (6) will necessarily be singular and using its pseudoinverse would mean *deleting* all but $n$ moment conditions. Therefore, we cannot simply use infinite or even too many moment conditions in GMM or OWGMM.

# 3 Methodology

We next present our approach. We start by motivating it using a new reformulation of OWGMM.

## 3.1 Reformulating OWGMM

Let us start by reinterpreting OWGMM. Consider the vector space $\mathcal{V}$ of real-valued functions $f$ of $Z$ under the usual operations. Note that, for any $\theta$, $\psi_n(f; \theta)$ is a linear operator on $\mathcal{V}$ and

$$\mathcal{C}_\theta(f, h) = \frac{1}{n} \sum_{i=1}^n f(Z_i) h(Z_i) (Y_i - g(X_i; \theta))^2$$

is a bilinear form on $\mathcal{V}$. Now, given any subset $\mathcal{F} \subseteq \mathcal{V}$, consider the following objective function:

$$\Psi_n(\theta; \mathcal{F}, \tilde{\theta}) = \sup_{f \in \mathcal{F}} \psi_n(f; \theta) - \frac{1}{4} \mathcal{C}_{\tilde{\theta}}(f, f). \tag{7}$$

**Lemma 1.** *Let $\|v\|_{\tilde{\theta}}$ be the optimally-weighted norm as in Eq. (6) and let $\mathcal{F} = \mathrm{span}(f_1, \ldots, f_m)$. Then*

$$\|(\psi_n(f_1; \theta), \ldots, \psi_n(f_m; \theta))\|_{\tilde{\theta}}^2 = \Psi_n(\theta; \mathcal{F}, \tilde{\theta}).$$

**Corollary 1.** *An equivalent formulation of OWGMM is*

$$\hat{\theta}^{OWGMM} \in \underset{\theta \in \Theta}{\mathrm{argmin}} \, \Psi_n(\theta; \mathcal{F}, \tilde{\theta}). \tag{8}$$

In other words, Lemma 1 provides a variational formulation of the objective function of OWGMM and Corollary 1 provides a saddle-point formulation of the OWGMM estimate.

## 3.2 DeepGMM

In this section, we outline the details of our DeepGMM framework. Given our reformulation above in Eq. (8), our approach is to simply replace the set $\mathcal{F}$ with a more flexible set of functions. Namely we let $\mathcal{F} = \{f(z; \tau) : \tau \in \mathcal{T}\}$ be the class of all neural networks of a given architecture with varying weights $\tau$ (but *not* their span). Using a rich class of moment conditions allows us to learn correspondingly a rich $g_0$. We therefore similarly let $\mathcal{G} = \{g(x; \theta) : \theta \in \Theta\}$ be the class of all neural networks of a given architecture with varying weights $\theta$.

Given these choices, we let $\hat{\theta}^{\mathrm{DeepGMM}}$ be the minimizer in $\Theta$ of $\Psi_n(\theta; \mathcal{F}, \tilde{\theta})$ for any, potentially data-driven, choice $\tilde{\theta}$. We discuss choosing $\tilde{\theta}$ in Section 4. Since this is no longer closed form, we formulate our algorithm in terms of solving a smooth zero-sum game. Formally, our estimator is defined as:

$$\hat{\theta}^{\mathrm{DeepGMM}} \in \underset{\theta \in \Theta}{\mathrm{argmin}} \sup_{\tau \in \mathcal{T}} \, U_{\tilde{\theta}}(\theta, \tau) \tag{9}$$

$$\text{where} \quad U_{\tilde{\theta}}(\theta, \tau) = \frac{1}{n} \sum_{i=1}^n f(Z_i; \tau)(Y_i - g(X_i; \theta)) - \frac{1}{4n} \sum_{i=1}^n f^2(Z_i; \tau)(Y_i - g(X_i; \tilde{\theta}))^2.$$

Since evaluation is linear, for any $\tilde{\theta}$, the game's payoff function $U_{\tilde{\theta}}(\theta, \tau)$ is convex-concave in the functions $g(\cdot; \theta)$ and $f(\cdot; \tau)$, although it may not be convex-concave in $\theta$ and $\tau$ as is usually the case when we parametrize functions using neural networks. Solving Eq. (9) can be done with any of a variety of smooth game playing algorithms; we discuss our choice in Section 4. We note that AGMM [28] also formulates IV estimation as a smooth game objective, but without the last regularization term and with the adversary parametrized as a mixture over a finite fixed set of critic functions.[3] In our experiments, we found the regularization term to be crucial for solving the game, and we found the use of a flexible neural network critic to be crucial with high-dimensional instruments.

Notably, our approach has very few tuning parameters: only the models $\mathcal{F}$ and $\mathcal{G}$ (*i.e.*, the neural network architectures) and whatever parameters the optimization method uses. In Section 4 we discuss how to select these.

Finally, we highlight that *unlike* the case for OWGMM as in Lemma 1, our choice of $\mathcal{F}$ is *not* a linear subspace of $\mathcal{V}$. Indeed, per Lemma 1, replacing $\mathcal{F}$ with a high- or infinite-dimensional linear subspace simply corresponds to GMM with many or infinite moments, which fails as discussed in Section 2.1 (in particular, we would generically have $\Psi_n(\theta; \mathcal{F}, \tilde{\theta}) = \infty$ unhelpfully). Similarly, enumerating many moment conditions as generated by, say, many neural networks $f$ and plugging these into GMM, whether one-step or optimally weighted, will fail for the same reasons. Instead, our approach is to leverage our variational reformulation in Lemma 1 and replace the class of functions $\mathcal{F}$ with a rich (non-subspace) set in this new formulation, which is distinct from GMM and avoids these issues. In particular, as long as $\mathcal{F}$ has bounded complexity, even if its ambient dimension may be infinite, we can guarantee the consistency of our approach. Since the last layer in a network is a linear combination of the penultimate one, our choice of $\mathcal{F}$ can in some sense be thought of as a union over neural network weights of subspaces spanned by the penultimate layer of nodes.

### 3.3 Consistency

Before discussing practical considerations in implementing DeepGMM, we first turn to the theoretical question of what consistency guarantees we can provide about our method if we were to approximately solve Eq. (9). We phrase our results for generic bounded-complexity functional classes $\mathcal{F}, \mathcal{G}$; not necessarily neural networks.

Our main result depends on the following assumptions, which we discuss after stating the result.

**Assumption 1** (Identification). $\theta_0$ is the unique $\theta \in \Theta$ satisfying $\psi(f; \theta) = 0$ for all $f \in \mathcal{F}$.

**Assumption 2** (Bounded complexity). $\mathcal{F}$ and $\mathcal{G}$ have vanishing Rademacher complexities:

$$\frac{1}{2^n} \sum_{\xi \in \{-1,+1\}^n} \mathbb{E} \sup_{\tau \in \mathcal{T}} \frac{1}{n} \sum_{i=1}^{n} \xi_i f(Z_i; \tau) \to 0, \quad \frac{1}{2^n} \sum_{\xi \in \{-1,+1\}^n} \mathbb{E} \sup_{\theta \in \Theta} \frac{1}{n} \sum_{i=1}^{n} \xi_i g(X_i; \theta) \to 0.$$

**Assumption 3** (Absolutely star shaped). For every $f \in \mathcal{F}$ and $|\lambda| \leq 1$, we have $\lambda f \in \mathcal{F}$.

**Assumption 4** (Continuity). For any $x$, $g(x; \theta)$, $f(x; \tau)$ are continuous in $\theta, \tau$, respectively.

**Assumption 5** (Boundedness). $Y$, $\sup_{\theta \in \Theta} |g(X; \theta)|$, $\sup_{\tau \in \mathcal{T}} |f(Z; \tau)|$ are all bounded random variables.

**Theorem 2.** *Suppose Assumptions 1 to 5 hold. Let $\tilde{\theta}_n$ by any data-dependent sequence with a limit in probability. Let $\hat{\theta}_n, \hat{\tau}_n$ be any approximate equilibrium in the game Eq. (9), i.e.,*

$$\sup_{\tau \in \mathcal{T}} U_{\tilde{\theta}_n}(\hat{\theta}_n, \tau) - o_p(1) \leq U_{\tilde{\theta}_n}(\hat{\theta}_n, \hat{\tau}_n) \leq \inf_{\theta} U_{\tilde{\theta}_n}(\theta, \hat{\tau}_n) + o_p(1).$$

*Then $\hat{\theta}_n \to \theta_0$ in probability.*

Theorem 2 proves that approximately solving Eq. (9) (with eventually vanishing approximation error) guarantees the consistency of our method. We next discuss the assumptions we made.

Assumption 1 stipulates that the moment conditions given by $\mathcal{F}$ are sufficient to identify $\theta_0$. Note that, by linearity, the moment conditions given by $\mathcal{F}$ are the same as those given by the subspace $\mathrm{span}(\mathcal{F})$ so we are actually successfully controlling many or infinite moment conditions, perhaps making the assumption defensible. If we do not assume Assumption 1, the arguments in Theorem 2 easily extend to showing instead that we approach *some* identified $\theta$ that satisfies all moment conditions. In particular this means that if we parametrize $f$ and $g$ via neural networks where we can permute the parameter vector $\theta$ and obtain an identical function, our result still holds. We formalize this by the following alternative assumption and lemma.

**Assumption 6** (Identification of $g$). Let $\Theta_0 = \{\theta \in \Theta : \psi(f; \theta) = 0 \ \forall f \in \mathcal{F}\}$. Then for any $\theta_1, \theta_2 \in \Theta_0$ the functions $g(\cdot; \theta_1)$ and $g(\cdot; \theta_2)$ are identical.

**Lemma 2.** *Suppose Assumptions 2 to 6 hold. Let $\hat{\theta}_n, \hat{\tau}_n$ be any approximate equilibrium in the game Eq. (9), i.e.,*

$$\sup_{\tau \in \mathcal{T}} U_{\tilde{\theta}_n}(\hat{\theta}_n, \tau) - o_p(1) \leq U_{\tilde{\theta}_n}(\hat{\theta}_n, \hat{\tau}_n) \leq \inf_{\theta} U_{\tilde{\theta}_n}(\theta, \hat{\tau}_n) + o_p(1).$$

*Then* $\inf_{\theta \in \Theta_0} \|\hat{\theta}_n - \theta\| \to 0$ *in probability.*

Assumption 2 provides that $\mathcal{F}$ and $\mathcal{G}$, although potentially infinite and even of infinite ambient dimension, have limited complexity. Rademacher complexity is one way to measure function class complexity [5]. Given a bound (envelope) as in Assumption 5, this complexity can also be reduced to other combinatorial complexity measures such VC- or pseudo-dimension via chaining [31]. [6] studied such combinatorial complexity measures of neural networks.

Assumption 3 is needed to ensure that, for any $\theta$ with $\psi(f; \theta) > 0$ for some $f$, there also exists an $f$ such that $\psi(f; \theta) > \frac{1}{4} C_{\tilde{\theta}}(f, f)$. It trivially holds for neural networks by considering their last layer. Assumption 4 similarly holds trivially and helps ensure that the moment conditions cannot simultaneously arbitrarily approach zero far from their true zero point at $\theta_0$. Assumption 5 is a purely technical assumption that can likely be relaxed to require only nice (sub-Gaussian) tail behavior. Its latter two requirements can nonetheless be guaranteed by either bounding weights (equivalently, using weight decay) or applying a bounded activation at the output. We do not find doing this is necessary in practice.

# 4 Practical Considerations in Implementing DeepGMM

**Solving the Smooth Zero-Sum Game.** In order to solve Eq. (9), we turn to the literature on solving smooth games, which has grown significantly with the recent surge of interest in generative adversarial networks (GANs). In our experiments we use the OAdam algorithm of [15]. For our game objective, we found this algorithm to be more stable than standard alternating descent steps using SGD or Adam.

Using first-order iterative algorithms for solving Eq. (9) enables us to effectively handle very large datasets. In particular, we implement DeepGMM using PyTorch, which efficiently provides gradients for use in our descent algorithms [30]. As we see in Section 5, this allows us to handle very large datasets with high-dimensional features and instruments where other methods fail.

**Choosing $\tilde{\theta}$.** In Eq. (9), we let $\tilde{\theta}$ be any potentially data-driven choice. Since the hope is that $\tilde{\theta} \approx \theta_0$, one possible choice is just the solution $\hat{\theta}^{\text{DeepGMM}}$ for another choice of $\tilde{\theta}$. We can recurse this many times over. In practice, to simulate many such iterations on $\tilde{\theta}$, we continually update $\tilde{\theta}$ as the previous $\theta$ iterate over steps of our game-playing algorithm. Note that $\tilde{\theta}$ is nonetheless treated as "constant" and does not enter into the gradient of $\theta$. That is, the second term of $U$ in Eq. (9) has zero partial derivative in $\theta$. Given this approach we can interpret $\tilde{\theta}$ in the premise of Theorem 2 as the final $\tilde{\theta}$ at convergence, since Theorem 2 allows $\tilde{\theta}$ to be fully data-driven.

**Hyperparameter Optimization.** The only parameters of our algorithm are the neural network architectures for $\mathcal{F}$ and $\mathcal{G}$ and the optimization algorithm parameters (*e.g.*, learning rate). To tune these parameters, we suggest the following general approach. We form a validation surrogate $\hat{\Psi}_n$ for our variational objective in Eq. (7) by taking instead averages on a validation data set and by replacing $\mathcal{F}$ with the pool of all iterates $f$ encountered in the learning algorithm for all hyperparameter choice. We then choose the parameters that maximize this validation surrogate $\hat{\Psi}_n$. We discuss this process in more detail in Appendix B.1.

**Early Stopping.** We further suggest to use $\hat{\Psi}_n$ to facilitate early stopping for the learning algorithm. Specifically, we periodically evaluate our iterate $\theta$ using $\hat{\Psi}_n$ and return the best evaluated iterate.

# 5 Experiments

In this section, we compare DeepGMM against a wide set of baselines for IV estimation. Our implementation of DeepGMM is publicly available at `https://github.com/CausalML/DeepGMM`.

We evaluate the various methods on two groups of scenarios: one where $X, Z$ are both low-dimensional and one where $X$, $Z$, or both are high-dimensional images. In the high-dimensional scenarios, we use a convolutional architecture in all methods that employ a neural network to accommodate the images. We evaluate performance of an estimated $\hat{g}$ by MSE against the true $g_0$.

More specifically, we use the following baselines:

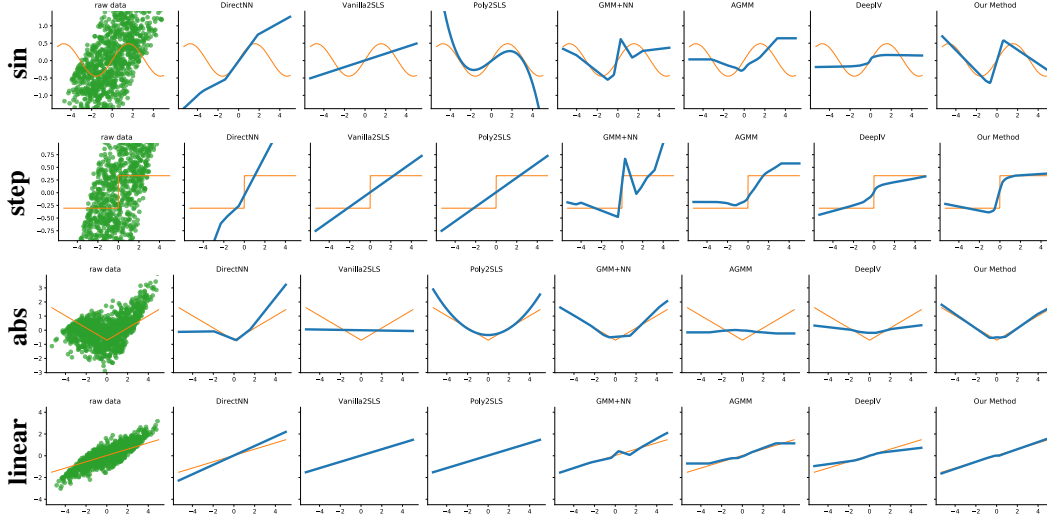

Figure 2: Low-dimensional scenarios (Section 5.1). Estimated $\hat{g}$ in blue; true response $g_0$ in orange.

| Scenario | DirectNN | Vanilla2SLS | Poly2SLS | GMM+NN | AGMM | DeepIV | Our Method |
|---|---|---|---|---|---|---|---|
| **sin** | $.26 \pm .00$ | $.09 \pm .00$ | $.04 \pm .00$ | $.08 \pm .00$ | $.11 \pm .01$ | $.06 \pm .00$ | $.02 \pm .00$ |
| **step** | $.21 \pm .00$ | $.03 \pm .00$ | $.03 \pm .00$ | $.06 \pm .00$ | $.06 \pm .01$ | $.03 \pm .00$ | $.01 \pm .00$ |
| **abs** | $.21 \pm .00$ | $.23 \pm .00$ | $.04 \pm .00$ | $.14 \pm .02$ | $.17 \pm .03$ | $.10 \pm .00$ | $.03 \pm .01$ |
| **linear** | $.09 \pm .00$ | $.00 \pm .00$ | $.00 \pm .00$ | $.06 \pm .01$ | $.03 \pm .00$ | $.04 \pm .00$ | $.01 \pm .00$ |

Table 1: Low-dimensional scenarios: Test MSE averaged across ten runs with standard errors.

| Scenario | DirectNN | Vanilla2SLS | Ridge2SLS | GMM+NN | AGMM | DeepIV | Our Method |
|---|---|---|---|---|---|---|---|
| **MNIST$_z$** | $.25 \pm .02$ | $.23 \pm .00$ | $.23 \pm .00$ | $.27 \pm .01$ | $-$ | $.11 \pm .00$ | $.07 \pm .02$ |
| **MNIST$_x$** | $.28 \pm .03$ | $> 1000$ | $.19 \pm .00$ | $.19 \pm .00$ | $-$ | $-$ | $.15 \pm .02$ |
| **MNIST$_{x,z}$** | $.24 \pm .01$ | $> 1000$ | $.39 \pm .00$ | $.25 \pm .01$ | $-$ | $-$ | $.14 \pm .02$ |

Table 2: High-dimensional scenarios: Test MSE averaged across ten runs with standard errors.

1. DirectNN: Predicts $Y$ from $X$ using a neural network with standard least squares loss.
2. Vanilla2SLS: Standard two-stage least squares on raw $X, Z$.
3. Poly2SLS: Both $X$ and $Z$ are expanded via polynomial features, and then 2SLS is done via ridge regressions at each stage. The regularization parameters as well polynomial degrees are picked via cross-validation at each stage.
4. GMM+NN: Here, we combine OWGMM with a neural network $g(x; \theta)$. We solve Eq. (5) over network weights $\theta$ using Adam. We employ optimal weighting, Eq. (6), by iterated GMM [20]. We are not aware of any prior work that uses OWGMM to train neural networks.
5. AGMM [28]: Uses the publicly available implementation[4] of the Adversarial Generalized Method of Moments, which performs no-regret learning on the one-step GMM objective Eq. (5) with norm $\| \cdot \|_\infty$ and an additional jitter step on the moment conditions after each epoch.
6. DeepIV [21]: We use the latest implementation that was released as part of the econML package.[5]

Note that GMM+NN relies on being provided moment conditions. When $Z$ is low-dimensional, we follow AGMM [28] and expand $Z$ via RBF kernels around 10 centroids returned from a Gaussian Mixture model applied to the $Z$ data. When $Z$ is high-dimensional, we use the moment conditions given by each of its components.[6]

## 5.1 Low-dimensional scenarios

In this first group of scenarios, we study the case when both the instrument as well as treatment is low-dimensional. Similar to [28], we generated data via the following process:

$$Y = g_0(X) + e + \delta \qquad\qquad X = Z_1 + e + \gamma$$
$$Z \sim \text{Uniform}([-3,3]^2) \qquad e \sim \mathcal{N}(0,1), \ \ \gamma, \delta \sim \mathcal{N}(0,0.1)$$

In other words, only the first instrument has an effect on $X$, and $e$ is the confounder breaking independence of $X$ and the residual $Y - g_0(X)$. We keep this data generating process fixed, but vary the true response function $g_0$ between the following cases:

**sin**: $g_0(x) = \sin(x)$     **step**: $g_0(x) = \mathbb{1}_{\{x \geq 0\}}$     **abs**: $g_0(x) = |x|$     **linear**: $g_0(x) = x$

We sample $n = 2000$ points for train, validation, and test sets each. To avoid numerical issues, we standardize the observed $Y$ values by removing the mean and scaling to unit variance. Hyperparameters used for our method in these scenarios are described in Appendix B.2. We plot the results in Fig. 2. The left column shows the sampled $Y$ plotted against $X$, with the true $g_0$ in orange. The other columns show in blue the estimated $\hat{g}$ using various methods. Table 1 shows the corresponding MSE over the test set.

First we note that in each case there is sufficient confounding that the DirectNN regression fails badly and a method that can use the IV information to remove confounding is necessary.

Our next substantive observation is that our method performs competitively across scenarios, attaining the *lowest MSE* in each (except linear where are beat just slightly and only by methods that use a linear model). At the same time, other methods employing neural networks perform well in some scenarios and less well in others. Therefore we conclude that in the low dimensional setting, our method is able to adapt to the scenario and compete with best tuned methods for the scenario.

Overall, we also found that GMM+NN performed well (but not as well as our method). In some sense GMM+NN is a novel method; we are not aware of previous work using (OW)GMM to train a neural network. Whereas GMM+NN needs to be provided moment conditions, our method can be understood as improving further on this by learning the best moment condition over a large class using optimal weighting. AGMM performed similarly well to GMM+NN, which uses the same moment conditions. Aside from the heuristic jitter step implemented in the AGMM code, it is equivalent to one-step GMM, Eq. (5), with $\| \cdot \|_\infty$ vector norm in place of the standard $\| \cdot \|_2$ norm. Its worse performance than our method perhaps also be explained by this change and by its lack of optimal weighting.

In the experiments, the other NN-based method, DeepIV, was consistently outperformed by Poly2SLS across scenarios. This may be related to the computational difficulty of its two-stage procedure, or possibly due to sensitivity of the second stage to errors in the density fitting in the first stage. Notably this is despite the fact that the neural-network-parametrized Gaussian mixture model fit in the first stage is correctly specified, so DeepIV's poorer performance cannot be attributed to the infamous "forbidden regression" issue. Therefore we might expect that, in more complex scenarios where the first-stage is not well specified, DeepIV could be at even more of a disadvantage. In the next section, we also discuss its limitations with high-dimensional $X$.

## 5.2 High-dimensional scenarios

We now move on to scenarios based on the MNIST dataset [26] in order to test our method's ability to deal with structured, high-dimensional $X$ and $Z$ variables. For this group of scenarios, we use same data generating process as in Section 5.1 and fix the response function $g_0$ to be **abs**, but map $Z$, $X$, or both $X$ and $Z$ to MNIST images. Let the output of Section 5.1 be $X^{\text{low}}, Z^{\text{low}}$ and $\pi(x) = \text{round}(\min(\max(1.5x+5, 0), 9))$ be a transformation function that maps inputs to an integer between 0 and 9, and let $\text{RandomImage}(d)$ be a function that selects a random MNIST image from the digit class $d$. The images are $28 \times 28 = 784$-dimensional. The scenarios are then given as:

- **MNIST$_Z$**: $X \leftarrow X^{\text{low}}$, $Z \leftarrow \text{RandomImage}(\pi(Z_1^{\text{low}}))$.

- **MNIST$_X$**: $X \leftarrow \text{RandomImage}(\pi(X^{\text{low}}))$, $Z \leftarrow Z^{\text{low}}$.

- **MNIST$_{X,Z}$**: $X \leftarrow \text{RandomImage}(\pi(X^{\text{low}}))$, $Z \leftarrow \text{RandomImage}(\pi(Z_1^{\text{low}}))$.

We sampled 20000 points for the training, validation, and test sets and ran each method 10 times with different random seeds. Hyperparameters used for our method in these scenarios are described in Appendix B.2. We report the averaged MSEs in Table 2. We failed to run the AGMM code on any of these scenarios, as it crashed and returned overflow errors. Similarly, the DeepIV code produced nan outcomes on any scenario with a high-dimensional $X$. Furthermore, because of the size of the examples, we were similarly not able to run Poly2SLS. Instead, we present Vanilla2SLS and Ridge2SLS, where the latter is Poly2SLS with fixed linear degree. Vanilla2SLS failed to produce reasonable numbers for high-dimensional $X$ because the first-stage regression is ill-posed.

Again, we found that our method performed competitively across scenarios, achieving the lowest MSE in each scenario. In the **MNIST$_Z$** setting, our method had better MSE than DeepIV. In the **MNIST$_X$** and **MNIST$_{X,Z}$** scenarios, it handily outperformed all other methods. Even if DeepIV had run on these scenarios, it would be at great disadvantage since it models the conditional distribution over images using a Gaussian mixture. This can perhaps be improved using richer conditional density models like [12, 22], but the forbidden regression issue remains nonetheless. Overall, these results highlights our method's ability to adapt not only to each low-dimensional scenario but also to high-dimensional scenarios, whether the features, instrument, or both are high-dimensional, where other methods break. Aside from our method's competitive performance, our algorithm was tractable and was able to run on these large-scale examples where other algorithms broke computationally.

## 6 Conclusions

**Other related literature and future work.** We believe that our approach can also benefit other applications where moment-based models and GMM is used [7, 18, 19]. Moreover, notice that while DeepGMM is related to GANs [16], the adversarial game that we play is structurally quite different. In some senses, the linear part of our payoff function is similar to that of the Wasserstein GAN [4]; therefore our optimization problem might benefit from a similar approaches to approximating the sup player as employed by WGANs. Another related line of work is in methods for learning conditional moment models, either in the context of IV regression or more generally, that are statistically efficient [1, 8–11]. This line of work is different in focus than ours; they focus on methods that are statistically efficient, whereas we focus on leveraging work on deep learning and smooth game optimization to deal with complex high-dimensional instruments and/or treatment. However an important direction for future work would be to investigate the possible efficiency of DeepGMM or efficient modifications thereof. Finally, there has been some prior work connecting GANs and GMM in the context of image generation [32], so another potential avenue of work would be to leverage some of the methodology developed there for our problem of IV regression.

**Conclusions.** We presented DeepGMM as a way to deal with IV analysis with high-dimensional variables and complex relationships. The method was based on a new variational reformulation of GMM with optimal weights with the aim of handling many moments and was formulated as the solution to a smooth zero-sum game. Our empirical experiments showed that the method is able to adapt to a variety of scenarios, competing with the best tuned method in low dimensional settings and performing well in high dimensional settings where even recent methods break.

## Acknowledgements

This material is based upon work supported by the National Science Foundation under Grant No. 1846210.

## Footnotes

[2]This assumption that a finite number of moment conditions uniquely identifies $\theta$ is perhaps too strong when $\theta$ is very complex, and it easily gives statistically efficient methods for estimating $\theta$ if true. However assuming this is difficult to avoid in practice.

[3] In their code they also include a jitter step where these critic functions are updated, however this step is heuristic and is not considered in their theoretical analysis.

[4] https://github.com/vsyrgkanis/adversarial_gmm

[5] https://github.com/microsoft/EconML

[6] That is, we use $f_i(Z) = Z_i$ for $i = 1, \ldots, \dim(Z)$.

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
