[Supplementary Material]

# A Omitted Proofs

*Proof of Lemma 1.* For ease of notation in this proof we will use $\|\cdot\|$ as shorthand for $\|\cdot\|_{\tilde{\theta}}$. First note that since $\|v\|^2 = v^T C_{\tilde{\theta}}^{-1} v$, the associated dual norm is $\|v\|_*^2 = v^T C_{\tilde{\theta}} v$. Next, define as shorthand $\boldsymbol{\psi}$ as shorthand for $(\psi_n(f_1; \theta), \ldots, \psi_n(f_m; \theta))$. It follows from the definition of the dual norm that $\|\boldsymbol{\psi}\| = \sup_{\|v\|_* \leq 1} v^T \boldsymbol{\psi}$. Therefore we have:

$$\|\boldsymbol{\psi}\|^2 = \sup_{\|v\|_* \leq \|\boldsymbol{\psi}\|} v^T \boldsymbol{\psi}$$
$$= \sup_{v^T C_{\tilde{\theta}} v \leq \|\boldsymbol{\psi}\|^2} v^T \boldsymbol{\psi}$$

The Lagrangian of this optimization problem is given by:

$$\mathcal{L}(v, \lambda) = v^T \boldsymbol{\psi} + \lambda(v^T C_{\tilde{\theta}} v - \|\boldsymbol{\psi}\|^2)$$

Taking the derivative of this with respect to $v$ shows us that when $\lambda < 0$, this quantity is maximized by $v = -\frac{1}{2\lambda} C_{\tilde{\theta}}^{-1} \boldsymbol{\psi}$. In addition we clearly have strong duality for this problem by Slater's condition whenever $\|\boldsymbol{\psi}\| > 0$ (since in this case $v = 0$ is a feasible interior point). This therefore gives us the following dual formulation for $\|\boldsymbol{\psi}\|^2$:

$$\|\boldsymbol{\psi}\|^2 = \inf_{\lambda < 0} -\frac{1}{2\lambda} \|\boldsymbol{\psi}\|^2 + \lambda(\frac{1}{4\lambda^2} \|\boldsymbol{\psi}\|^2 - \|\boldsymbol{\psi}\|^2)$$
$$= \inf_{\lambda < 0} -\frac{1}{4\lambda} \|\boldsymbol{\psi}\|^2 - \lambda \|\boldsymbol{\psi}\|^2$$

Taking derivative with respect to $\lambda$ we can see that this is minimized by setting $\lambda = -\frac{1}{2}$. Given this and strong duality, we know it must be the case that $\|\boldsymbol{\psi}\|^2 = \sup_v \mathcal{L}(v, -\frac{1}{2}) = \sup_v v^T \psi - \frac{1}{2} v^T C_{\tilde{\theta}} v + \frac{1}{2} \|\boldsymbol{\psi}\|^2$. Rearranging terms and doing a change of variables $v \leftarrow 2v$ gives us the identity:

$$\|\boldsymbol{\psi}\|^2 = \sup_v v^T \boldsymbol{\psi} - \frac{1}{4} v^T C_{\tilde{\theta}} v$$

Finally, we can note that any vector $v \in \mathbb{R}^m$ corresponds to some $f \in \text{span}(\mathcal{F})$, such that $f = \sum_i v_i f_i$, and according to this notation we have $v^T \boldsymbol{\psi} = \psi_n(f; \theta)$ and $v^T C_{\tilde{\theta}} v = \mathcal{C}_{\tilde{\theta}}(f, f)$. Therefore our required result follows directly from the previous identity. $\square$

*Proof of Theorem 2.* Define $m(\theta, \tau, \tilde{\theta}) = f(Z; \tau)(Y - g(X; \theta) - \frac{1}{4} f(Z; \tau)^2 (Y - g(X; \tilde{\theta}))^2$, $M(\theta) = \sup_{\tau \in \mathcal{T}} \mathbb{E}[m(\theta, \tau, \tilde{\theta})]$, and $M_n(\theta) = \sup_{\tau \in \mathcal{T}} \mathbb{E}_n[m(\theta, \tau, \tilde{\theta}_n)]$, where $\mathbb{E}_n$ refers to the empirical measure (average over the $n$ data points) and $\tilde{\theta}_n \to_p \tilde{\theta}$. We will proceed by proving the following three conditions, and then proving our results in terms of these conditions:

1. $\sup_\theta |M_n(\theta) - M(\theta)| \to_p 0$

2. for every $\delta > 0$ we have $\inf_{d(\theta, \theta_0) \geq \delta} M(\theta) > M(\theta_0)$

3. $M_n(\hat{\theta}_n) \leq M_n(\theta_0) + o_p(1)$

We will proceed by proving these conditions one by one. For the first, we can derive the inequality:

$$\sup_\theta |M_n(\theta) - M(\theta)|$$
$$= \sup_\theta \left| \sup_\tau \mathbb{E}_n[m(\theta, \tau, \tilde{\theta}_n)] - \sup_\tau \mathbb{E}[m(\theta, \tau, \tilde{\theta})] \right|$$
$$\leq \sup_{\theta, \tau} \left| \mathbb{E}_n[m(\theta, \tau, \tilde{\theta}_n)] - \mathbb{E}[m(\theta, \tau, \tilde{\theta})] \right|$$
$$\leq \sup_{\theta, \tau} \left| \mathbb{E}_n[m(\theta, \tau, \tilde{\theta}_n)] - \mathbb{E}[m(\theta, \tau, \tilde{\theta}_n)] \right| + \sup_{\theta, \tau} \left| \mathbb{E}[m(\theta, \tau, \tilde{\theta}_n)] - \mathbb{E}[m(\theta, \tau, \tilde{\theta})] \right|$$
$$\leq \sup_{\theta_1, \theta_2, \tau} |\mathbb{E}_n[m(\theta_1, \tau, \theta_2)] - \mathbb{E}[m(\theta_1, \tau, \theta_2)]| + \sup_{\theta, \tau} \left| \mathbb{E}[m(\theta, \tau, \tilde{\theta}_n)] - \mathbb{E}[m(\theta, \tau, \tilde{\theta})] \right|$$

Next we will bound these two terms separately, which we will term $B_1$ and $B_2$. For the first term, we can derive the following bound, where $\epsilon_i$ are iid Rademacher random variables, $m_i(\theta, \tau, \tilde{\theta}_n) = f(Z_i; \tau)(Y_i - g(X_i; \theta) - \frac{1}{4}f(Z_i; \tau)^2(Y_i - g(X_i; \tilde{\theta}))^2$, and $m_i'(\theta, \tau, \tilde{\theta}_n')$ are shadow variables:

$$\mathbb{E}[B_1] = \mathbb{E}\left[\sup_{\theta_1, \theta_2, \tau} \left| \frac{1}{n} \sum_i m_i(\theta_1, \tau, \theta_2) - \mathbb{E}[m_i'(\theta_1, \tau, \theta_2')] \right| \right]$$

$$\leq \mathbb{E}\left[\sup_{\theta_1, \theta_2, \tau} \left| \frac{1}{n} \sum_i m_i(\theta_1, \tau, \theta_2) - m_i'(\theta_1, \tau, \theta_2') \right| \right]$$

$$= \mathbb{E}\left[\sup_{\theta_1, \theta_2, \tau} \left| \frac{1}{n} \sum_i \epsilon_i(m_i(\theta_1, \tau, \theta_2) - m_i'(\theta_1, \tau, \theta_2')) \right| \right]$$

$$\leq 2\mathbb{E}\left[\sup_{\theta_1, \theta_2, \tau} \left| \frac{1}{n} \sum_i \epsilon_i m_i(\theta_1, \tau, \theta_2) \right| \right]$$

$$\leq 2\mathbb{E}\left[\sup_{\theta, \tau} \left| \frac{1}{n} \sum_i \epsilon_i f(Z_i; \tau)(Y_i - g(X_i; \theta)) \right| \right]$$

$$+ \frac{1}{2}\mathbb{E}\left[\sup_{\theta, \tau} \left| \frac{1}{n} \sum_i \epsilon_i f(Z_i; \tau)^2(Y_i - g(X_i; \theta))^2 \right| \right]$$

$$\leq \mathbb{E}\left[\sup_{\theta, \tau} \left| \frac{1}{n} \sum_i \epsilon_i f(Z_i; \tau)^2 \right| \right] + \mathbb{E}\left[\sup_{\theta, \tau} \left| \frac{1}{n} \sum_i \epsilon_i(Y_i - g(X_i; \theta)^2 \right| \right]$$

$$+ \frac{1}{4}\mathbb{E}\left[\sup_{\theta, \tau} \left| \frac{1}{n} \sum_i \epsilon_i f(Z_i; \tau)^4 \right| \right] + \frac{1}{4}\mathbb{E}\left[\sup_{\theta, \tau} \left| \frac{1}{n} \sum_i \epsilon_i(Y_i - g(X_i; \theta))^4 \right| \right]$$

Note that in the final inequality we apply the inequality $xy \leq 0.5(x^2 + y^2)$. Now given Assumption 5, the functions that map $(f(Z_i; \tau)$ and $g(X_i; \theta))$ to the summands in each term are Lipschitz. Now for any function class $\mathcal{F}$ and $L$-Lipschitz function $\phi$ we have $\mathcal{R}_n(\phi \circ \mathcal{F}) \leq L\mathcal{R}_n(\mathcal{F})$, where $\mathcal{R}_n(\mathcal{F})$ is the Rademacher complexity of class $\mathcal{F}$ [25, Thm. 4.12]. Therefore we have

$$\mathbb{E}[B_1] \leq L(\mathcal{R}_n(\mathcal{G}) + \mathcal{R}_n(\mathcal{F})),$$

for some constant $L$. Thus given Assumption 2 it must be case that $\mathbb{E}[B_1] \to 0$. Now let $B_1'$ be some recalculation of $B_1$ where we are allowed to edit the $i$'th $X$, $Z$, and $Y$ values. Then given Assumption 5 we can derive the following bounded differences inequality:

$$\sup_{X_{1:n}, Z_{1:n}, Y_{1:n}, X_i', Z_i', Y_i'} |B_1 - B_1'| \leq \sup_{\theta_1, \theta_2, \tau, X_{1:n}, Z_{1:n}, Y_{1:n}, X_i', Z_i', Y_i'} \left| \frac{1}{n}(m_i(\theta_1, \tau, \theta_2) - m_i'(\theta_1, \tau, \theta_2)) \right|$$

$$\leq \frac{c}{n}$$

for some constant $c$. Therefore from McDiarmid's Inequality we have $P(|B_1 - \mathbb{E}[B_1]| \geq \epsilon) \leq 2\exp\left(-\frac{2n\epsilon^2}{c^2}\right)$. Putting this and the previous result for $\mathbb{E}[B_1]$ together we get $B_1 \to_p 0$.

Next, define $\omega_n = \left| (Y - g(X; \tilde{\theta}_n))^2 - (Y - g(X; \tilde{\theta}))^2 \right|$. Recall that from the premise of the theorem we have $\tilde{\theta}_n \to_p \tilde{\theta}$. Then by Slutsky's Theorem, the Continuous Mapping Theorem, and Assumption 4 we have $\omega_n = o_p(1)$. Given this we can bound $B_2$ as follows:

$$B_2 = \sup_{\theta, \tau} \left| \mathbb{E}[m(\theta, \tau, \tilde{\theta}_n)] - \mathbb{E}[m(\theta, \tau, \tilde{\theta})] \right|$$

$$= \frac{1}{4} \sup_{\theta, \tau} \left| \mathbb{E}[f(Z; \tau)^2(Y - g(X; \tilde{\theta}_n))^2] - \mathbb{E}[f(Z; \tau)^2(Y - g(X; \tilde{\theta}))^2] \right|$$

$$\leq \frac{1}{4} \sup_{\theta, \tau} \left| \mathbb{E}[f(Z; \tau)^2(Y - g(X; \tilde{\theta}))^2] - \mathbb{E}[f(Z; \tau)^2(Y - g(X; \tilde{\theta}))^2] \right| + \frac{1}{4} \sup_{\tau} \left| \mathbb{E}[f(Z; \tau)^2 \omega_n] \right|$$

$$= \frac{1}{4} \sup_{\tau} \left| \mathbb{E}[f(Z; \tau)^2 \omega_n] \right|$$

Now we know from Assumption 5 that $f(Z; \tau)$ is uniformly bounded, so it follows that $B_2 \leq \frac{b}{4} \mathbb{E}[|\omega_n|]$ for some constant $b$. Next we can note, again based on our boundedness assumption, that $\omega_n$ is uniformly bounded. Therefore it follows from the Lebesgue Dominated Convergence Theorem that $\mathbb{E}[|\omega_n|] \to 0$. Thus we know that both $B_1$ and $B_2$ converge, so we have proven the first of the three conditions, that $\sup_\theta |M_n(\theta) - M(\theta)|$ converges in probability to zero.

For the second condition we will first prove that $M(\theta_0)$ is the unique minimizer of $M(\theta)$. Clearly by Assumptions 1 and 3 we have that $\theta_0$ is the unique minimizer of $\sup_\tau \mathbb{E}[f(Z; \tau)(Y - g(X; \theta)]$, since it sets this quantity to zero, and by these assumptions any other value of $\theta$ must have at least one $\tau$ that can be played in response that makes this expectation strictly positive. Now we can see that $M(\theta_0) = 0$ also, since $M(\theta_0) = \sup_\tau -\frac{1}{4} f(Z; \tau)^2 (Y - g(X; \tilde{\theta}))^2$, and the inside of the supremum is clearly non-positive but can be set to zero using the zero function for $f$, which is allowed given Assumption 3. Furthermore, for any other $\theta' \neq \theta_0$, let $f'$ be some function in $\mathcal{F}$ such that $\mathbb{E}[f(Z)(Y - g(X; \theta')] > 0$. If we have $\mathbb{E}[f'(Z)^2 (Y - g(X; \tilde{\theta}))^2] = 0$ then it follows immediately that $M(\theta') > 0$. Otherwise, consider the function $\lambda f'$ for arbitrary $0 < \lambda < 1$. Since by Assumption 3 this function is also contained in $\mathcal{F}$, it follows that:

$$M(\theta') = \sup_{f \in \mathcal{F}} \mathbb{E}[f(Z)(Y - g(X; \theta')) - \frac{1}{4} f(Z)^2 (Y - g(X; \tilde{\theta}))^2]$$

$$\geq \lambda \mathbb{E}[f'(Z)(Y - g(X; \theta'))] - \frac{\lambda^2}{4} \mathbb{E}[f'(Z)^2 (Y - g(X; \tilde{\theta}))^2]$$

This expression is a quadratic in $\lambda$ that is clearly positive when $\lambda$ is sufficiently small, so therefore it still follows that $M(\theta') > 0$.

Given this, we will prove the second condition by contradiction. If this were false, then for some $\delta > 0$ we would have that $\inf_{\theta \in B(\theta_0, \delta)} M(\theta) = M(\theta_0)$, where $B(\theta_0, \delta) = \{\theta \mid d(\theta, \theta_0) \geq \delta\}$. This is because from Assumption 1 we know $\theta_0$ is the unique minimizer of $M(\theta)$. Given this there must exist some sequence $(\theta_1, \theta_2, \ldots)$ in $B(\theta_0, \delta)$ satisfying $M(\theta_n) \to M(\theta_0)$. Now by construction $B(\theta_0, \delta)$ is closed, and the corresponding limit parameters $\theta^* = \lim_{n \to \infty} \theta_n \in B(\theta_0, \delta)$ must satisfy $M(\theta^*) = M(\theta_0)$, since given Assumption 4 $M(\theta)$ is clearly a continuous function of $\theta$ so we can swap function application and limit. However $d(\theta^*, \theta_0) \geq \delta > 0$, so $\theta^* \neq \theta_0$. This contradicts the fact that $\theta_0$ is the unique minimizer of $M(\theta)$, so we have proven the second condition.

Finally, for the third condition we will use the fact that by assumption $\hat{\theta}_n$ satisfies the approximate equilibrium condition:

$$\sup_{\tau \in \mathcal{T}} \mathbb{E}_n m(\hat{\theta}_n, \tau, \tilde{\theta}_n) - o_p(1) \leq \mathbb{E}_n m(\hat{\theta}_n, \hat{\tau}_n, \tilde{\theta}_n) \leq \inf_\theta \mathbb{E}_n m(\theta, \hat{\tau}_n, \tilde{\theta}_n) + o_p(1)$$

Now by definition $M_n(\hat{\theta}_n) = \sup_{\tau \in \mathcal{T}} \mathbb{E}_n m(\hat{\theta}_n, \tau, \tilde{\theta}_n)$. Therefore,

$$\inf_\theta \mathbb{E}_n m(\theta, \hat{\tau}_n, \tilde{\theta}_n) \leq \inf_\theta \sup_\tau \mathbb{E}_n m(\theta, \tau, \tilde{\theta}_n) = \inf_\theta M_n(\theta) \leq M_n(\theta_0).$$

Thus we have

$$M_n(\hat{\theta}_n) - o_p(1) \leq \mathbb{E}_n m(\hat{\theta}_n, \hat{\tau}_n, \tilde{\theta}_n) \leq M_n(\theta_0) + o_p(1).$$

At this point we have proven all three conditions stated at the start of the proof. For the final part we can first note that from the first and third conditions it easily follows that $M_n(\hat{\theta}_n) \leq M(\theta_0) + o_p(1)$, since $|M_n(\theta_0) - M(\theta_0)| \to_p 0$. Therefore we have:

$$M(\hat{\theta}_n) - M(\theta_0) \leq M(\hat{\theta}_n) - M_n(\hat{\theta}_n) + o_p(1)$$

$$\leq \sup_\theta \left| M(\hat{\theta}) - M_n(\hat{\theta}) \right| + o_p(1)$$

$$\leq o_p(1)$$

Next, define $\eta(\delta) = \inf_{d(\theta, \theta_0) \geq \delta} M(\theta) - M(\theta_0)$. Now by definition of $\eta$ we know that whenever $d(\hat{\theta}_n, \theta) \geq \delta$ we have $M(\hat{\theta}_n) - M(\theta_0) \geq \eta(\delta)$. Therefore $\mathbb{P}[d(\hat{\theta}_n, \theta) \geq \delta] \leq \mathbb{P}[M(\hat{\theta}_n) - M(\theta_0) \geq \eta(\delta)]$. Now since for every $\delta > 0$ we have $\eta(\delta) > 0$ from the second condition, and we know $M(\hat{\theta}_n) - M(\theta_0) = o_p(1)$, we have that for every $\delta > 0$ the RHS probability converges to zero. Thus $d(\hat{\theta}_n, \theta_0) = o_p(1)$, so we can conclude that $\hat{\theta}_n \to_p \theta_0$. $\qquad \square$

*Proof of Lemma* 2. We prove this lemma by redefining the parameter space $\Theta$ such that it satisfies Assumptions 1 to 5, and then appealing to the proof of Theorem 2. We define the pseudo parameter space $\Theta^*$ as

$$\Theta^* = \{\Theta_0\} \cup \{\theta \in \Theta : \theta \notin \Theta_0\}.$$

In other words, we group together all elements of $\Theta_0$ as a single element that we will refer to as $\theta^*$. Note that by definition of $\Theta_0$ the set of functions $g(\cdot;\theta)$ induced by $\theta \in \Theta'$ is the same as for the original parameter space, and that the function $g(\cdot;\theta^*)$ is well-defined given Assumption 6. Next we define the metric $d'$ for the new parameter space in terms of the metric of the original space $d$ as

$$d'(\theta_1,\theta_2) = \min\{d(\theta_1,\theta_2), \inf_{\theta_0 \in \Theta_0} d(\theta_1,\theta_0) + \inf_{\theta_0 \in \Theta_0} d(\theta_2,\theta_0)\}$$

Next we justify that $d'$ is a metric. First of all it is trivial from this definition that $d'(\theta,\theta) = 0$ for every $\theta \in \Theta'$, and also that non-negativity and symmetry are maintained. Finally it is easy to verify that this definition still satisfies the triangle inequality, given that $d$ is a metric and satisfies the triangle inequality itself.

In order to verify that we still have continuity, consider $\theta_1$ and $\theta_2$ such that $d'(\theta_1,\theta_2) < \epsilon$. In the case that $d(\theta_1,\theta_2) = d'(\theta_1,\theta_2)$, continuity follows trivially from Assumption 4. That is, it is trivial to construct an $\epsilon,\delta$ argument in this case. If this isn't the case, there must exist $\theta_1^*,\theta_2^* \in \Theta_0$ such that $d(\theta_1,\theta_1^*) + d(\theta_2,\theta_2^*) = \epsilon$. In addition for any $x$ we have $|g(x;\theta_1) - g(x;\theta_2)| \leq |g(x;\theta_1) - g(x;\theta_1^*)| + |g(x;\theta_2^*) - g(x;\theta_2)|$, given Assumption 6. Thus continuity still follows and it is trivial to construct a formal $\epsilon,\delta$ argument given Assumption 4.

Given that all the other assumptions only depend on the space of functions $g(\cdot;\theta)$ not on the parameter space itself, they are unaffected via switching from $\Theta$ to $\Theta^*$. Also, by construction using $\Theta^*$ instead of $\Theta$ means we satisfy Assumption 1. Thus we satisfy Assumptions 1 to 5, so we can appeal to Theorem 2 to argue that $d'(\hat{\theta}_n - \theta^*) \to 0$ in probability. Finally it follows from the definition of $d'$ that $d'(\theta,\theta^*) = \inf_{\theta_0 \in \Theta_0} d(\theta,\theta_0)$, which gives us our final result that $\inf_{\theta_0 \in \Theta_0} d(\hat{\theta}_n,\theta_0) \to 0$ in probability. □

# B  Additional Methodology Details

## B.1  Hyperparameter Optimization Procedure

We provide more details here about the hyperparameter optimization procedure described in Section 4. Let $m$ be the total number of hyperparameter choices under consider. Then for each candidate set of hyperparameters $\gamma_i \in \{\gamma_1,\ldots,\gamma_m\}$ we run our learning algorithm for a fixed number of epochs using $\gamma_i$, training it on our train partition. Every $k_{\text{eval}}$ epochs we save the current parameters $\hat{\tau}$ and $\hat{\theta}$ at that epoch. This gives, for each hyperparameter choice $\gamma_i$, a finite set of $f$ functions $\hat{\mathcal{F}}_i$, and a finite set of $\theta$ values $\hat{\Theta}_i$.

Now, define the set of functions $\hat{\mathcal{F}} = \cup_{i=1}^m \hat{\mathcal{F}}_i$. We define our approximation of our variational objective as

$$\hat{\Psi}_n(\theta) = \Psi_n(\theta; \hat{\mathcal{F}}, \theta),$$

where $\Psi_n$ is as defined in Eq. (7). Note that this means for every $\theta$ we wish to evaluate we choose to approximate $\tilde{\theta}$ using that value of $\theta$.

Given this, we finally choose the set of hyperparameters $\gamma_i$ whose corresponding trajectory of parameter values $\hat{\Theta}_i$ minimizes the objective function $\min_{\theta \in \hat{\Theta}_i} \hat{\Psi}_n(\theta)$, calculated on the validation data. Note that this objective function is meant to approximate the value of the variational objective we would have obtained if we performed learning with that set of hyperparameters using early stopping.

Note that in practice, since we only ever calculate $\hat{\Psi}_n$ on the validation data, and we only ever use $\Theta_i$ in optimizing the above objective function on the validation data, instead of saving the actual parameter values $\hat{\tau}$ and $\hat{\theta}$ we can instead save vectors $f(Z_{\text{val}},\tau)$ and $g(X_{\text{val}},\theta)$, where $Z_{\text{val}}$ and $X_{\text{val}}$ are the vectors of $Z$ and $X$ values respectively in our validation data. This makes our methodology tractable when working with very complex deep neural networks.

| scenario | $g$ model | $f$ model | $g$ learning rates | $\lambda_f$ |
|---|---|---|---|---|
| low-dimensional | FCNN $(20, 3)$ | FCNN $(20)$ | $(5 * 10^{-4}, 2 * 10^{-4}, 1 * 10^{-3})$ | 5.0 |
| **MNIST$_z$** | FCNN $(200, 200)$ | CNN | $(2 * 10^{-5}, 5 * 10^{-5}, 1 * 10^{-4})$ | 5.0 |
| **MNIST$_x$** | CNN | FCNN $(20)$ | $(1 * 10^{-6}, 2 * 10^{-6}, 5 * 10^{-6})$ | 1000.0 |
| **MNIST$_{x,z}$** | CNN | CNN | $(1 * 10^{-6}, 2 * 10^{-6}, 5 * 10^{-6})$ | 5.0 |

Table 3: Hyperparameters used in our experiments.

## B.2 Hyperparameter Details

We describe here the specific hyperparameter choices used in all our experiments. In all scenarios we parametrized the $f$ and $g$ networks either as fully connected neural networks (FCNN) with leaky ReLU activations with various numbers and sizes of hidden layers, or using a fixed deep convolutional neural network (CNN) architecture designed to perform well with non-causal inference on the MNIST data. We refer readers to our code release for exact details on our CNN construction. In addition in all cases we performed the hyperparameter optimization procedure described above over a range of learning rates. Specifically, in every scenario we explore a range of learning rates for $g$, and compute the $f$ learning rate as $\text{lr}_f = \lambda_f \text{lr}_g$, where $\lambda_f$ is chosen separately for each scenario.

We summarize our choices for each scenario in Table 3. Note that we made the same hyperparameter choices in all low-dimensional scenarios. In the case of FCNN models, we list the hidden layer sizes in parentheses.

### B.2.1 Low-dimensional Scenarios

In the low dimensional scenarios we parametrized both $f$ and $g$ as fully-connected neural networks, using a single hidden layer of size 20 for $f$, and two hidden layers of sizes 20 and 3 for $g$. We performed hyperparameter optimization over a range of learning rates, searching over the learning rates $[5 * 10^{-4}, 2 * 10^{-4}, 1 * 10^{-3}]$ for $g$, and in each instance multiplying learning rate by 5 for $f$.

### B.2.2 MNIST$_z$ Secario

In this scenario we parametrized

## C One-Step GMM Using All Square Integrable Moments

We describe here the equivalence of performing one-step GMM using all square integrable functions of the instruments, and non-causal least squares. Since we are considering an infinite collection of moment functions we only consider the infinity norm (*i.e.* $\|v\| = \|v\|_\infty$.) In addition, the measure we use for integration is the empirical measure. That is, we use all instruments $f$ such that $\sum_{i=1}^n f(Z_i)^2 = 1$. Letting $B_1$ denote this set, $\mathbb{E}_n$ denote expectation with respect to the empricial measure, and otherwise using the same notation as in Section 2.1, we obtain:

$$
\begin{aligned}
\|\psi_n(f_i; \theta), \dots \|^2 &= \sup_{f \in B_1} \mathbb{E}_n[f(Z)(Y - g(X; \theta))]^2 \\
&= \sup_{f \in B_1} \mathbb{E}_n[f(Z)\mathbb{E}_n[Y - g(X; \theta) \mid Z]]^2 \\
&= \left( \frac{\mathbb{E}_n[\mathbb{E}_n[Y - g(X; \theta) \mid Z]^2]}{\mathbb{E}_n[\mathbb{E}_n[Y - g(X; \theta) \mid Z]^2]^{1/2}} \right)^2 \\
&= \mathbb{E}_n[\mathbb{E}_n[Y - g(X; \theta) \mid Z]^2] \\
&= \mathbb{E}_n[(Y - g(X; \theta))^2],
\end{aligned}
$$

where the third line follows because $f$ takes it supremum at $f(Z) = \mathbb{E}_n[Y - g(X; \theta) \mid Z]$ by Cauchy-Schwarz, and the final line follows under the additional assumption that $Z$ is a continuous random variable, which implies that $Y - g(Z; \theta)$ has zero conditional variance under the empirical measure given any $Z_i$. Thus we observe that using the infinity norm and under the additional assumption

that $Z$ is continuous, one-step GMM using these moment functions is equivalent to non-causal least squares, since they both involve picking $\theta$ to minimize the same loss function.