[Reviews · NeurIPS 2019]

Reviewer 1



Originality: This work builds on recent work on adapting deep networks for use with instrumental variables (DeepIV [Hartford et al 2017] & Adversarial GMM (AGMM) [Lewis & Syrgkanis 2018]) but adapts the optimally weighted GMM [Hansen 1982] (OWGMM) for the task. AGMM is probably most similar in that it is also an adversarial loss, but the variational reformulation presented in this paper results in a far simpler algorithm. Quality: I thought this was great paper. The variational reformulation of OWGMM leads to a far simpler objective function that neatly leverages the explosion of recent work in adversarial learning (GANs, etc.) by replacing a large number of moment conditions with a single adversarial network. That said, given that the method appears useful in practice, I would have liked to see more detailed experiments on the practical considerations. For example - I don't follow the GAN literature closely, but my impression is that adversarial losses can be finicky to train - is this a problem in practice or does OAdam lead to reliable convergence (what do typical learning curves look like?)? How well does the validation loss correlate with the loss under an interventional distribution? How sensitive is the procedure to the choice of \tilde{\theta}? Finally - I found the discussion of the low dimensional experiments very misleading. GMM+NN is described as having "performed well" and outperforming AGMM but it only did so significantly on 1 / 4 experiments (it's beaten on linear and step and abs don't have significant differences). Then the subsequent paragraph attributes DeepIV's performance relative to Poly2SLS the "forbidden regression". But it significantly outperformed all other neural network methods aside from this new approach. The discussion is particularly misleading because the "forbidden regression" argument in [Angrist & Pischke 2008] doesn't consider cross validation. This is a 1-D Gaussian that DeepIV is approximating with a mixture of Gaussians - assuming you're validating correctly there's no reason for misspecification to be an issue. A far more likely explanation is Poly2SLS is properly tuned (the polynomial degrees were selected by cross-validation) while DeepIV wasn't fully trained. I'm not at all surprised that tuned Poly2SLS would perform well on a 1D problem that's well-approximated by a polynomial. Clarity: I found the paper to be generally very well-written and clear, with only the experimental section missing important details. In particular: - what were the chosen hyper parameters for the deep networks? Number of layers / units per layer / regularization used / etc? - how did you choose hyper parameters for your baselines? You mention cross-validation for Poly2SLS - did you do anything similar for AGMM and DeepIV? - the DGP description looks like it doesn't line up with the figure 2. If X = 0.5 Z + 0.5 e then I'd expect X to have a range \approx [-2, 2], so [-4, 4] seems wrong. - How is test MSE computed? Is it on the intervention distribution (i.e. is drawn from an unconfounded distribution at test time)? Is it MSE relative to the true response?

Reviewer 2



Originality: The theoretical contributions are completely novel. In spite of the similarity of their approach to AGMM, the formulation of the problem is based on a different objective function that allows optimal reweighing, instead of the unweighted moment conditions. Quality: The algorithmic, theoretical and empirical contributions are sound. Though the evaluation is not extensive, it is convincing. Clarity: The clarity and organization of the paper can be improved 1) It might be helpful to use different notations for the unweighted norm and weighted norm. In particular, lemma 1 would read better without having to refer back to the definition of the weighted norm. 2) Line 102 about equivalence to non-causal linear regression requires justification or a reference. 3) Line 79 is not meaningful. There ought to be some relationship between $m$ and complexity of $\theta$ 4) Theorem 2. It says that $\tilde{\theta}_n$ has a limit. Can this be any limit? or it should be $\theta_0$? If not, its quite counter intuitive and requires further explanation. 5) In particular, AGMM paper also suggests a way of learning the moment functions via deep networks. What precisely makes DeepGMM needs to be emphasized. 6) Line 190. it says that $\tilde{\theta}$ does not enter the gradient of $\theta$. Wouldn't that mean that the optimum of $\theta$ does not depend on $\tilde{\theta}$? Perhaps you just mean that $\tilde{\theta}$ should be treated as constants when taking gradient w.r.t \theta. It can still be a part of the gradient. 7) Line 224. Please elaborate on "When Z is high-dimensional, we use the moment conditions given by each of its components". 8) I could not understand the data for high dimensional case. Seems like $X$ is sometimes and image and sometimes a number. Furthermore, defining $g_0$ as abs would mean that one is taking absolute value of an image. Is that meaningful? Significance: The paper is an important contribution to the field of causality research. And likely to be used considering the performance of the algorithm. --- Post rebuttal comments: The authors responded adequately to most of my concerns, but they did not clarify comment 8 in my review. Furthermore, I agree with the issues pointed out by the other reviewers on the experimental section. I have lowered my score to reflect that.

Reviewer 3



I found the paper interesting. In particular, I liked the variational formulation of optimally-weighted generalized method of moments. This formulation is likely useful when the number of moments are large and inverting the covariance matrix is computationally difficult. I also commend the attempt at proving consistency when both the causal response function g and moment function f are parameterized by neural networks. The empirical results seem promising compared to alternatives, particularly in the high-dimensional case. What would make this paper stronger is addressing some gaps from Lemma 1 to the proposed DeepGMM. First, the identification assumption is not true for neural network G's (e.g. a permutation of hidden units yields the same output). I know that identification is a standard assumption in GMM literature, and is not valid here, but discussing a bit more about the richness of \mathcal{F} needed to obtain correct \theta would be helpful (maybe as part of lines 158-163). Perhaps a larger issue than the identification condition is understanding if the -1/4 C(f, f) term is needed in the neural network f case. From my reading of the proof of Theorem 2, this term is not needed to prove consistency, since by Assumptions 2 and 5, the f's are already well-behaved. I would assume that one might obtain similar experimental results by other controls on f (perhaps by spectral normalization or gradient penalties), since -1/4 C(f, f) is only well-motivated in the "finite" f case. (This is somewhat of a minor comment, as there is some motivation from optimal weighting, but I'm not sure that the optimal weighting perspective makes sense in the infinite f case). The experimental results look promising, but I think it is missing some key details. First, it is worth understanding what the architectures for g and f were. Depending the architecture for g, it could be that the moment conditions for the baseline results in Sections 5.1 and 5.2 were not sufficiently "rich" to recover theta (I'm thinking mostly about Poly2SLS/Ridge2SLS) here. Second, it would nice to know how many hyperparameters (as defined in B.1) were used to train the network. Third, it would be nice to know other details, such as the learning rate, the number of training steps, batch size, etc. Other minor comments: 1.) It would be good to cite the paper "Kernel Instrumental Variable Regression" https://arxiv.org/abs/1906.00232. (This came out at ICML, so I'm not sure you were aware of the work during submission. 2.) The paper was on the whole well written, but I found a couple typos - line 144 "implementating" -> "implementing" - line 289, double citation of reference 15 - line 377 I think the equation should be "- 1/4 v'Cv" and not "+ 1/4 v'Cv" lines 381/382... I think you're missing some close parens here 3.) On related and future work, there is some work on learning implicit generative models using GMM-line techniques (e.g. "Learning Implicit Generative Models with the Method of Learned Moments" https://arxiv.org/abs/1806.11006), and it would be interesting to see a variant of the variational formulation used here. I hope the authors address the above comments, because I think the paper is promising and has some nice ideas.

[Author Response · NeurIPS 2019]

We would like to thank the reviewers for their useful, detailed feedback! We will update the paper with the suggested minor revisions regarding typos and presentation improvements, and respond to individual reviewers comments below.

Reviewer #1:

1. Corollary 1 doesn't actually imply the consistency proven in Theorem 1, because it only applies to a finite number of moment conditions, whereas in Theorem 1 we deal with an infinite number of moment conditions and a function space defined by a neural network architecture, which is why proving consistency there is important and necessary.

2. You're correct that training the adversarial loss function can be finicky, especially if done naively. We find that in practice when we include the $-\frac{1}{4}C$ term inspired by Lemma 1 learning is very stable in particular in the low dimensional scenarios, and other tweaks like OAdam and early stopping using our validation scheme seem to improve this further.

3. We agree that some of the prose discussing GMM+NN results is confusing vis-a-vis results in Table 1. We will alter this discussion accordingly to more clearly reflect the numbers.

4. We acknowledge your point about lack of misspecification in DeepIV's first stage in the low-dim scenarios, and so forbidden regression might not be the right explanation for its performance there. However, the DeepIV second stage network does have the capacity to fit the problems better than a polynomial. Regarding "full training": we used an existing implementation of DeepIV.

5. The network architectures and hyperparams were mistakenly omitted from the supplement and will be added. Note our code is public. For baseline methods we used the official implementations provided by the authors.

6. Thanks for catching, we apologize for the typo. Both 0.5's should be 1's: $X = Z_1 + e + \gamma$.

7. Yes, test MSE is w.r.t. the "true response" $g_0$ over the $X$ population, as mentioned in 2nd paragraph of Sec 5.

Reviewer #3:

1. Good idea. We will use the notation $\|x\|_C = x^T C^{-1} x$.

2. We will add a proof in the appendix. This is just self-duality of $L_2$.

3. Regarding line 79, our comment is simply pointing out that standard GMM usually starts from the assumption that the moment conditions specify the problem. Indeed, the assumption that a finite number of moment conditions identify theta is very strong (too strong) when theta is complex because it easily gives us statistically efficient methods for estimating theta if true. We will clarify.

4. Yes; the limit of $\tilde{\theta}_n$ can be anything. This may seem counterintuitive, but one way of understanding this is that having the "correct" limit corresponds to using the optimally weighted norm, whereas having some other limit corresponds to some non-optimal norm, but using an non-optimal norm for GMM just means that estimator won't be statistically efficient, not that it won't be consistent.

5. All theory in AGMM applies to an *a priori* fixed finite collection of moment conditions; the only "learning [of] moment functions" in AGMM is in a heuristic jitter step added in the experiments. The significant differences between DeepGMM and AGMM, other than the drastic difference in performance, are that our method is inspired by the statistically efficient optimally weighted GMM with a corresponding regularization term which AGMM lacks, that we directly optimize the neural net critic $f$, and that we do learn a critic from an infinite collection rather than using a finite ensemble of critics. We will update this to make it clearer.

6. Yes; as line 190 says, what we mean is $\tilde{\theta}$ is treated as constant. The second term of $U$ in Equation (9) has zero partial derivative in $\theta$; so $\tilde{\theta}$ does not appear in the $\theta$ gradient. We will rephrase to make this clearer.

7. This means we use $f_i(Z) = Z_i$ for $i = 1, \ldots, 784$. We'll add this clarification.

Reviewer #4:

1. Identification assumption for neural nets: as referenced in line 161, we can easily relax identification and instead converge to *some* $\theta$ satisfying the moment conditions (will clarify this simple extension in the proof). Moreover, this immediately gives that even if there are redundancies in that two $\theta$s give the same function $g$ (e.g., permuting hidden layers), if $g \in G$ is identified (i.e., all identified thetas give rise to the same function $g$), we will obtain *some* parameterization $\theta$ of the unique $g$. It's a good point so we will add this discussion.

2. Regarding the $-\frac{1}{4}C(f, f)$ term, we found that other regularizers/controls on f do not perform well as they induce suboptimal weighting that ignores the covariance of the moment conditions for different f, whereas our new regularizer, as you write, is motivated by optimal weighting. This a key driver of our improved performance over, e.g., AGMM.

3. The network architectures and learning hyperparams were mistakenly omitted from the supplement and will be added. Note our code is public. Re "sufficiently rich" moments for Poly2SLS: since its degree is variable, Poly2SLS can be thought of as a sieve as in Newey and Powell [23], giving universal consistency.

4. We will cite the recent paper you reference on kernelized 2SLS. It wasn't on our radar (first appearance online June 2019). We will also cite the Ravuri et al. paper re future work. Thanks.

[Meta-Review · NeurIPS 2019]

This paper received good scores overall, with a substantial disparity in the initial scores that was later reduced after rebuttal and discussions. There is consensus that it makes novel contributions in using moment-matching techniques for causal inference with instrumental variables. The main weakness of this paper are the weak experiments and lack of details about the model.